# Genetic structure of black soldier flies in northern Iran

Afrooz Boukan[1], Jamasb Nozari[1]*, Nazanin Naseri Karimi[2], Fahimeh Talebzadeh[2], Katayoun Pahlavan Yali[1], Mohammad Ali Oshaghi[2]*

1 Department of Plant Protection, Faculty of Agriculture and Natural Resources, University of Tehran, Karaj, Iran, 2 Departement of Medical Entomology and Vector Control, School of Public Health, Tehran University of Medical Sciences, Tehran, Iran

* nozari@ut.ac.ir (JN); moshaghi@sina.tums.ac.ir (MAO)

## Abstract

### Background

The black soldier fly (BSF), *Hermetia illucens*, is known for nutrient-recycling through the bioconversion of organic waste into protein-rich insect larvae that can be processed into an animal feed ingredient. However, information on species distribution and its genetic structure in Iran is scarce.

### Methods and results

We directed a survey on the Caspian Sea coast, with a reconstructing demographic relationships study using two parts of mitochondrial cytochrome C oxidase 1 (COI) gene (barcode and 3' end regions) and nuclear internal transcribed spacer 2 (ITS2) to identify BSF's genetic diversity in retrospect to the global diversity and the potential origin of the Iranian BSF population. Larvae and adults were recovered from highly decomposed poultry manure, in May 2020. Sequence analysis of both regions of COI gene (about 1500 bp) revealed a single haplotype, identical to that of haplotype C, a worldwide commercial strain originated from Nearctic, Palearctic, or African biogeographic regions. However, the ITS2 locus was confirmed to be invariable across samples from diverse biogeographic regions.

### Conclusion

The results proved the presence of BSF in north of Iran. However, it is not possible to determine with certainty when and where this species first established in Iran, and they have likely been released to nature due to the existence of companies importing and breeding such flies. Due to heavy international trading, the introduction and settlement of this fly in the southern coasts of the country is highly suggested.

**Data Availability Statement:** The relevant sequence data for this study are publicly available from the NIH GenBank database (https://www.ncbi.nlm.nih.gov/genbank/) with the following

accession numbers: OQ803472-OQ803477, PQ032498, and PQ058287.

**Funding:** Tehran University of Medical Sciences.

**Competing interests:** The authors have declared that no competing interests exist.

## Introduction

*Hermetia illucens* (L.) (Diptera, Stratiomyidae) is known as the black soldier fly (BSF) and is a promising candidate for nutrient-recycling through the bioconversion of organic waste into biomass in the insect farming industry [1–3]. BSF is known as a dietary source of insect protein and lipids worldwide [4, 5]. Fat accumulates rapidly in BSF and reaches 20–40% [6] which can be used for biodiesel production [7]. The insect fat biomass has many valuable properties, such as diesel compatibility, high energy content and cetane number [8–10]. Farming BSF on various organic wastes has been suggested as an excellent opportunity for producing nutrient-rich animal organic fertilizer, feed, fuel, and bio-based products [11, 12]. At present there are several registered trademarks of the larvae, that are used as food in poultry and aquaculture farms [13–16, also see https://phoenixworm.com]. The larvae of this species are active decomposers of various rotting organic matter, including fruit and vegetable waste, agro-industrial by-products, organic leachates, corpses, and food waste [17–20]. Some larval by-products of this fly are also used for eradicating multidrug-resistant bacteria in aquaculture [21], however, these by-products are not currently being used in industrial aquaculture. BSF are considered to be a possible source of antimicrobial peptides (AMPs) for substitution of antibiotics in livestock farming, which is beneficial to human and animal welfare [22–24]. Currently, the genes related to nutrients and energy processing have been characterized in BSF [25]. BSF, as a sarcosaprophagous insect, has been used for estimation of postmortem interval (PMI) in forensic investigations [26]. From a public health perspective, BSF larvae are thought to prevent infestations by house flies, *Musca domestica* L. (Diptera: Muscidae), seeking oviposition sites that lead to reduction of the house fly population [27]. The larvae have six larval stages before pupation and adult females lay 500–1000 eggs [28]. Although BSF is a valuable insect in different aspects, its distribution in some regions of the world is not known.

Although this species is cosmopolitan and reported in many different biogeographical regions of the world [29], it is believed that the species originates in the New World [2, 30]. Therefore, the Old-World populations are considered non-indigenous, and thought to be mainly introduced by shipping [30]. In Asia, this species was first reported from Malaysia in the 1940s and later from China in 1960 [30]. However, information on species distribution and abundance, and its genetic structure, in Iran is scarce: the species appears to have been introduced to the country through international trade in the region over the last decades. It is known that many of the initially known BSF habitats outside the New World are near to the coast or on islands [31]. Even though BSF is being used by some industrial companies in Iran, however, the natural history and the genetic structure of the BSF are poorly known.

Studying of cytochrome oxidase subunit I gene of mitochondrial DNA (mtDNA-COI) has been used as an advanced systematic tool applied in evolutionary and population study on species taxonomy and delimitation [32]. This gene is one of the biggest genes in the metazoan mitochondrial genome and it is characterized by higher variability than the other mitochondria genes. It is also one of the most important molecular markers used for molecular taxonomy and systematic of living things and microorganisms [33–36]. Of different segments of the COI gene, DNA barcode region at 5' end as well as the 3' end of the gene are used vastly in molecular systematic in literature [37–42]. The barcoding region provides a common source of DNA sequence for taxonomy and identification of organisms [43], whereby scientists can compare living organisms. Moreover, the internal transcribed spacer II (ITS2) ribosomal DNA locus has previously been applied to enhance species identification of different insects including beetles [44], ticks [45], sand flies [46], and mosquitoes [34, 47, 48].

Here we carried out the first survey, across the Caspian Sea coast in northern Iran, to use molecular techniques for the BSF taxonomy. We investigated the morphological characteristics

of the field-collected flies during the 2020, and then determined their genetic structure, using two mitochondrial and a nuclear molecular marker: the 5' end of cytochrome oxidase I (COI), known as the barcode region, the 3' end of COI, and the ITS2.

## Material and methods

### Study area, sample collection, and species identification

The study was conducted along the Caspian Sea coast in Mazandaran (36.3994° N, 52.1912° E) and Gilan (37.1172° N, 49.5280° E), two Northern provinces of Iran. The areas investigated included Astara, Talesh, Asalem, Anzali, Rasht, Lahijan, Klachai in Gilan Province and Shirood, Ramsar, Tonekabon, Chalous, Amol, Marzango, Ghaemshahr, CheftKola, Jouybar (Sarvkola) in Mazandaran Province. Adults (n = 12) and larval (n>100) of the BSF were captured respectively with an insect net and forceps in autumn (n = 3 adults) (temperature 10°C and relative humidity (RH) 65%) and winter (n = 9 adults) (temperature 3°C and RH 60%), from a residential area in the Caspian region, and near decaying organic matter, particularly poultry manure and slaughterhouse waste. Some specimens were kept alive for cultivation, and a subset of specimens was stored in 70% ethanol at 4°C until analysis. The collected specimens were transferred to the Bio-systematic Entomology Laboratory of the University of Tehran for species identification. Morphological identification of specimens was performed using the key of de Carvalho and De Mello-Patiu [49]. To dissect the genitals, the two extremities of the abdomen were removed with a dissection needle and cleaned with 10% potassium hydroxide (KOH). Slides were then prepared from the genitals with Mount Canada balsam solution. The main characteristics of this species included two segmented cerci, wing pattern, naked eyes, and the pattern on the eyes (Fig 1). Illustrations were made using a ZEISS Standard Microscope and Optika SZM 1 stereomicroscope equipped with a Sony TM digital camera. Finally, the species diagnosis was confirmed by Dr. Martin Hauser, senior insect biosystematist, California Department of Food and Agriculture, Plant Pest Diagnostics Branch, USA. A subset of morphologically identified specimens was selected for molecular analysis in the Insect Molecular Biology Lab, Department of Biology of Vectors and Control of Diseases, Tehran University of Medical Sciences.

### Molecular analysis

Prior to DNA extraction, adult black solder flies were individually rinsed twice with 70% ethanol, centrifuged at 12,000 RPM for 2 min and left to dry in the air. One thorax of an individual fly was used for DNA extraction. The dried thorax was ground with sterile pestles in DNA lysis buffer. DNA extraction was performed using a commercial kit (Qiagen DNeasy Blood and Tissue kit, Hilden, Germany), according to the manufacturer's instructions. The quality and quantity of the extracted DNA was estimated using the spectrophotometer (Thermo Scientific™ NanoDrop™ One) and 1% agarose gel electrophoresis. From this specimen DNA, two parts of the mitochondrial protein-coding COI gene and the nuclear ITS2 region were amplified. The first amplicon was an 884-bp region of the 3' end of the COI gene, which was amplified by polymerase chain reaction (PCR), using the primers C1-J-2183: (5′-CAACATTTATTTTGA TTTTTGG-3′) and T1-2-n-3014 (5′-CCATTGCATCTGCCATATTA-3′) [32]. The second amplicon was a 709-bp region of 5' end of the COI gene, known as the COI barcode region [33], which was amplified using the universally conserved primer pairs LCO-1490 (5′-GGTC AACAAATCATAAAGATATTGG-3′) and HCO-2198 (5′-TAAACTTCAGGGTGACCAAAA AATCA-3′) [50]. The third amplicon was the ITS2 rDNA locus which was amplified and sequenced using the thermal cycling program and primers (5.8S: 5′-TGTGAACTGCA GGACACAT-3′ and 28S: 5′-TATGCTTAAATCAGGGGGT-3′) reported by Ståhls et al [29].

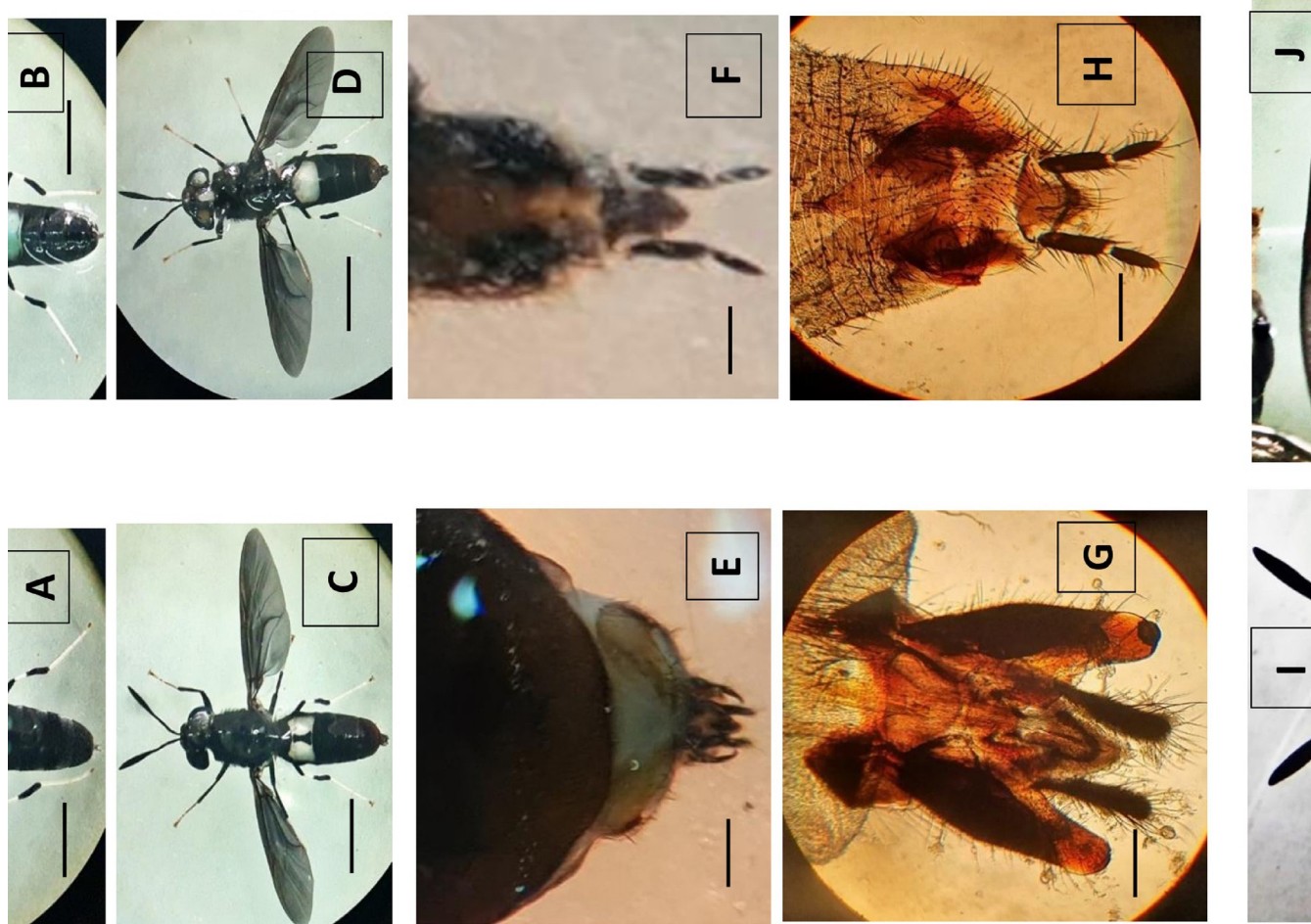

**Fig 1. Morphological characteristics of *Hermetia illucens* captured in northern Iran, 2020.** A, Dorsal view of female. B, Ventral view of female. C, Dorsal view of male. D. Ventral view of male. E & G, Tip of abdomen and genital capsule (10x), male. Scale represents 2mm. F & H, Tip of abdomen and Genital structure (10x), female, dorsal view. I, Dorsal view of the head. J, Wing.

All PCR reactions were conducted in a total volume of 25μL, using the Taq DNA Polymerase 2x Master Mix RED, Ampliqon (Denmark), with the following reagents: 1–2μL of DNA extract (10–25 ng), 12.5μL of Master mix, 1μL of each primer (10mM), and 8.5–9.5μL of sterile water. For both COI regions, the PCR thermal profile was 2 min at 94°C, followed by 5 cycles of 94°C (30s), 45°C (40s), and 72°C (60s), and then 35 cycles of 94°C (30s), 51°C (40s), and 72°C (60s), with a final extension at 72°C for 10 min. PCR products were separated by 1% agarose gel electrophoresis, followed by Green Viewer staining and imaging using a UV transilluminator. Successful amplicons were sequenced bidirectionally by the Genetic Codon Company, Tehran, Iran, using the same primers as for amplification.

## Analysis of DNA sequences

The quality of crude sequences was improved using the Chromas 2.6.5 program by removing areas of poor quality at each end of the sequences. The consensus of confident sequences was then examined using the NCBI (nucleotide collection) database. Multiple sequences were aligned using the Clustal Omega package [51]. The homology of DNA sequences generated in

this study to GenBank entries were measured using The Basic Local Alignment Search Tool (BLAST). For phylogenetic analysis, a representative barcode or 3' end sequence of the Iranian black soldier fly were compared with those barcode (n = 36) or 3' end (n = 17) publically-available DNA from Genbank databases, which represent different haplotypes [52] and/or world biogeographic regions, including Nearctic, Neotropical, Palearctic, East-Palearctic, Afrotropical, Oriental, and Australasia, as already described by Ståhls et al. [29] and Guilliet et al. [52] (Tables 1 and 2). Phylogenetic analysis was performed using the Neighbor-Joining method embedded in MEGA X [53] with bootstrap values being calculated from 1000 trees. The trees were drawn to scale, with branch lengths in the same units as those of the evolutionary distances used to infer the phylogenetic tree.

**Compliance with Ethical Standards.** This article does not contain any studies with human participants or animals performed by any of the authors. The protocols were conducted in this study followed the guidelines of the institutional ethical committee (University of Tehran).

## Results

Our morphological investigation revealed the presence of a field population of BSF in Lahijan county located in the Caspian coast region in northern Iran. A subset of sampled specimens from that population was randomly selected for molecular analysis, and the remaining specimens (larvae or adults) were kept alive and used for population rearing.

### COI 3'-end and barcode analyses

The PCR amplicons were produced successfully for both 5'- and 3'- parts of the COI gene. The COI dataset sequences for both regions were obtained for a subset (n = 5) of the field-collected specimens. After trimming, the sequences of about 628 bp in length for the 5' end and 840 bp in length for the 3'-end of the gene, respectively, were obtained and have been deposited in GenBank under accession numbers OQ803468-OQ803477. The A + T contents were 62.0 and 65.95% (mean 63.97%) respectively for 5'-end (barcode) and 3'-end regions. All sequences from this population were identical.

The sequence data were compared with a number of publically-available COI DNA barcodes and 5'-end regions from Genbank sequence databases. BLAST analysis showed that the barcode region of Iranian BSF specimens were identical to their counterparts in China, USA, Bhutan, Russia, Spain, France, and Netherlands from various biogeographical regions comprising East-Palearctic, Nearctic, Oriental, and Palearctic.

When the sequences of the 3'-end of the COI gene were compared with available data in GenBank or literature [52], the Iranian BSF population matched with various samples from USA, China, Taiwan, UK, France, Kenya, and Ghana. Further analysis revealed that the Iranian specimens were 100% identical to haplotype C (Table 2), which corresponds to globally common commercial strains [29, 52].

### Phylogenetic analysis

This analysis showed that the maximum sequence divergence among the haplotypes was 4.3% for 5' end (628 bp) and 5.83% for 3' end (840 bp) of COI gene. The Neighbor-Joining phylogenetic analysis for the COI barcode region could resolve the specimens into two main clades (Fig 1), each containing two subclades. The Iranian specimen (Palearctic region) was sister taxon with representative specimens from East-Palearctic, Palearctic, Oriental, and Nearctic regions. The Iranian specimen was associated with the commercial specimens (haplotype C) from China, Netherlands, Russia, Bhutan, Spain, France, and USA. Haplotype B was the closest

**Table 1. The sequence data of 5′ ends of the COI barcode region was used for phylogenetic analysis in this study.** DS: direct submission.

| Biogeographical Region | Haplotype* | Continent | Country | GenBank ID number | Reference |
|---|---|---|---|---|---|
| PALEARCTIC | C | Asia | Iran | OQ803468-OQ803472 | This study |
| ORIENTAL | C | ASIA | Bhutan | LR778160 | [29] |
| ORIENTAL | E | ASIA | Singapore | MT178485 | [29] |
| ORIENTAL | E | ASIA | Bhutan | LR778159 | [29] |
| ORIENTAL | U | ASIA | Thailand | LR778162 | [29] |
| ORIENTAL | A | ASIA | Malaysia | LR792223 | [29] |
| AFROTROPICAL | C | AFRICA | Kenya | LR778157 | [29] |
| AFROTROPICAL | G | AFRICA | Benin | MT151287 | [29] |
| AFROTROPICAL | A | AFRICA | La Reunion | MT151288 | [29] |
| AFROTROPICAL | U | AFRICA | South Africa | MT178507 | [29] |
| AFROTROPICAL | G | AFRICA | Ghana | LR778158 | [52] |
| PALEARCTIC | C | EUROPE | Spain | LR792260 | [29] |
| PALEARCTIC | A | EUROPE | Switzerland | LR778207 | [29] |
| PALEARCTIC | F | EUROPE | Switzerland | LR792261 | [29] |
| PALEARCTIC | U/C | EUROPE | France | MT178493 | [29] |
| PALEARCTIC | C | EUROPE | Netherlands | MT483933 | DS |
| PALEARCTIC | C | EUROPE | Russia | KY817115 | DS |
| EAST_PALEARCTIC | C | ASIA | China | NC_035232 | DS |
| EAST_PALEARCTIC | F | ASIA | S. Korea | FJ794401 | [52] |
| EAST_PALEARCTIC | B | ASIA | S. Korea | HQ541188 | [54] |
| EAST_PALEARCTIC | A | ASIA | S. Korea | HQ541228 | [54] |
| NEOTROPICAL | D | AMERICA | Mexico | LR778208 | [29] |
| NEOTROPICAL | H | AMERICA | Bolivia | LR778193 | [52] |
| NEOTROPICAL | A | AMERICA | Bolivia | LR792195 | [29] |
| NEOTROPICAL | U | AMERICA | Colombia | LR778204 | [29] |
| NEOTROPICAL | U | AMERICA | Peru | LR778213 | [29] |
| NEOTROPICAL | U | AMERICA | Peru | LR792231 | [52] |
| NEOTROPICAL | U | AMERICA | Brazil | LR778206 | [29] |
| NEOTROPICAL | H | AMERICA | Brazil | LR792236 | [52] |
| NEARCTIC | U/C | AMERICA | USA | MT178490 | [29] |
| NEARCTIC | U | AMERICA | USA | MT178458 | [29] |
| NEARCTIC | E | AMERICA | Canada | KM967419 | [55] |
| AUSTRALASIA | U | AUSTRALIA | Australia | LR792227 | [29] |
| AUSTRALASIA | E | AUSTRALIA | Australia | MT178508 | [29] |
| AUSTRALASIA | A | AUSTRALIA | Australia | LR792225 | [29] |
| AUSTRALASIA | E | AUSTRALIA | Australia | LR792256 | [29] |

*: Based on ten major COI haplotypes (A-J, and U: unknown) suggested by Guilliet et al. [52]. The Blast search revealed 100% similarity between the Iranian specimen with the GenBank entries NC_035232, OQ803468, OQ690653, OP537078, KY817115, MT483939, MT483933, LR778160, MT178494, and MT483932 from various geographical regions of the world.

haplotype to haplotype C. Other haplotypes including A, D, E, G, and H were positioned in a separate clade comprising representatives from Neotropical, Nearctic, Afrotropical, Australasia, and Oriental regions while haplotype F samples were grouped in both main clades (Fig 2).

The phylogenetic analysis for the 3' end of COI gene showed similar topography with the tree inferred from the 5' end of the COI gene (Fig 2). Similarly, the Iranian BSF population

**Table 2. The sequence data of 3′ ends of the COI gene (840 bp) was used for phylogenetic analysis in this study.**

| Biogeographical Region | Haplotype* | Continent | Country | GenBank ID Number/code** | Reference |
|---|---|---|---|---|---|
| PALEARCTIC | C | Asia | Iran | OQ803473-OQ803477 | This study |
| AFROTROPICAL | C | AFRICA | Kenya | a7 | [52] |
| AFROTROPICAL | C | AFRICA | Kenya | a25 | [52] |
| AFROTROPICAL | C | AFRICA | Ghana | a2 | [52] |
| PALEARCTIC | A | EUROPE | France | a11 | [52] |
| PALEARCTIC | H | EUROPE | France | a39 | [52] |
| PALEARCTIC | J | EUROPE | France | a48 | [52] |
| PALEARCTIC | F | EUROPE | France | a52 | [52] |
| PALEARCTIC | C | EUROPE | France | a12 | [52] |
| PALEARCTIC | C | EUROPE | France | a10 | [52] |
| PALEARCTIC | H | EUROPE | Guyane | a16 | [52] |
| PALEARCTIC | C | EUROPE | UK | a12 | [52] |
| EAST_PALEARCTIC | C | ASIA | Taiwan | a8 | [52] |
| EAST_PALEARCTIC | C | ASIA | China | NC_035232 | [56] |
| NEOTROPICAL | D | AMERICA | Mexico | a15 | [52] |
| NEARCTIC | C | AMERICA | USA | Berkly female | [52] |

*: Based on seven major mitochondrial genome haplotypes (A-C-D-F-G-H-J) suggested by Guilliet et al 2022.

**: Adapted from Guilliet et al. [52].

was associated with the commercial specimens (haplotype C) from USA, China, Taiwan, UK, France, Kenya, and Ghana and altogether constituted a distinct clade far from other haplotypes. Other haplotypes including A, D, G, H, and J were positioned in a separate clade comprising representatives from Neotropical, Nearctic, Afrotropical, Australasia, and Oriental regions (Fig 3).

## ITS2 rDNA analyses

The PCR amplicons (about 453 bp) were produced successfully for ITS2 region. The ITS2 sequences were obtained for a subset (n = 5) of the field-collected specimens. After trimming, the sequences of about 433 bp in length were obtained and have been deposited in GenBank under accession numbers PQ032498 and PQ058287. The sequences of all five specimens were identical. The sequence data were compared with all publicly-available ITS2 DNA regions from GenBank sequence databases. This analysis showed that the ITS2 region of Iranian BSF specimens were identical to their counterparts from various biogeographical regions comprising Neotropical, Nearctic, Afrotropical, and Palearctic. Because the ITS2 rDNA sequences displayed no intraspecific variation among sequences, no phylogenetic analysis was performed for the ITS2 sequences.

## Discussion

Here in this study, we report the presence of the black soldier fly (BSF), *H. illucens*, in northern Iran based on morphological characteristics and molecular analyses of COI and ITS2 loci. This fly has great ecological and economic value, and the economic value and the ecological consequences of mass-producing this fly in an Iranian context would be worth investigating. Since only one report of Stratiomyidae was made in the Middle East in 2024 [57] the present study,

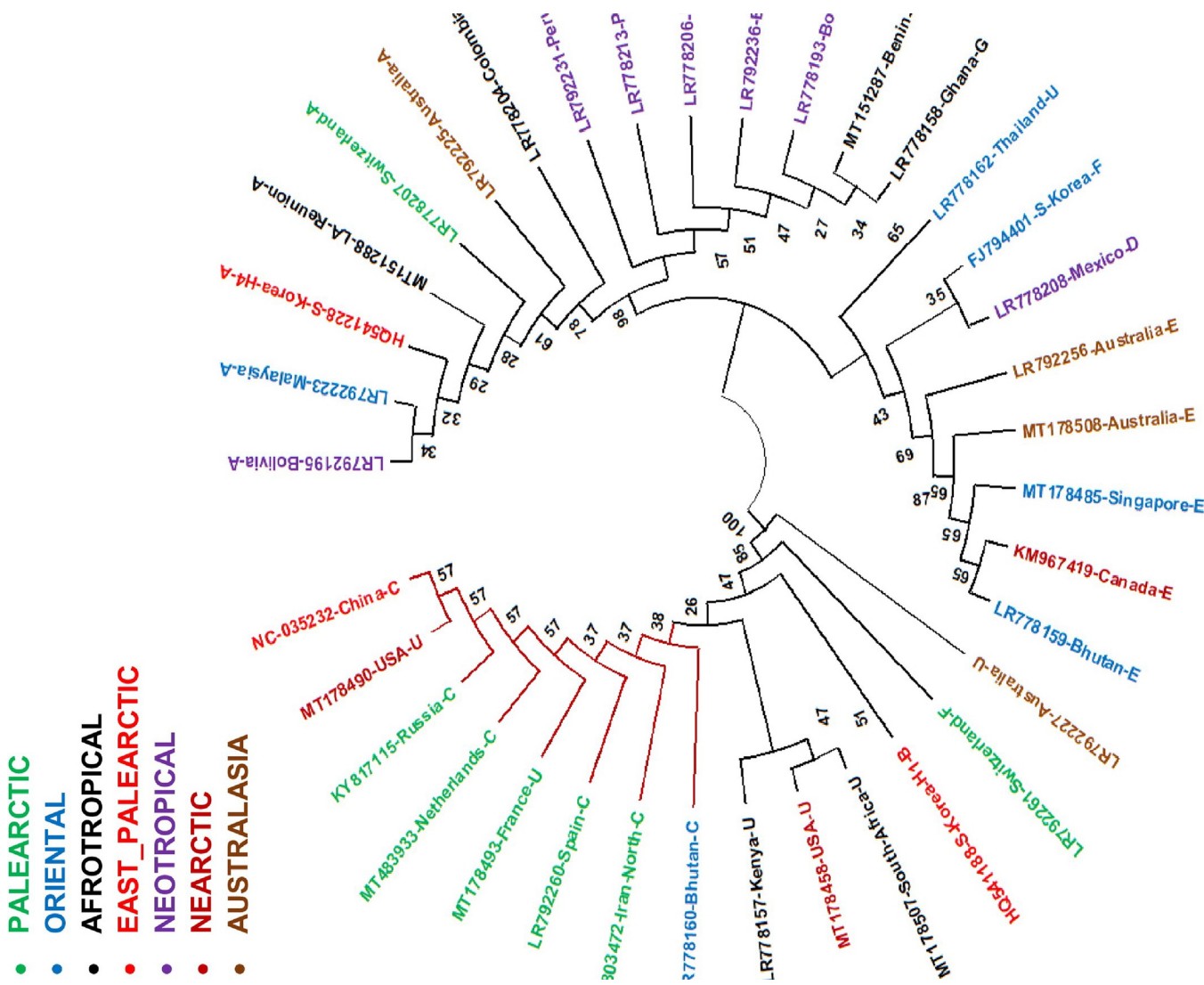

**Fig 2. The phylogenetic tree inferred by using Neighbor-Joining method based on 628 bp of the barcode regions of COI gene of *Hermetia illucens*.** The Iranian representative is labeled with red star. The other data are representatives of available sequences in GenBank and are labelled with country and haplotype (A-H, U represents unknown) [44]. The bootstrap values of more than 50 are shown on the internal nodes. The colors of the branches correspond to the biogeographical regions.

which is based on morphological and molecular diagnosis on the population of BSF caught from nature, is the second report of the entry and establishment of this species in the Middle East ecosystem. It remains unclear whether the investigated population can be considered naturalized in Northern Iran or is subject to recurrent yet temporary occurrences, possibly going back to flies escaping from BSF rearing facilities in the wider region. The finding of BSF in the region is important for people who are involved in agriculture, aquaculture, animal husbandry, and poultry-keeping, as well as public health and waste management. In northern Iran the available lands and water are limited and, furthermore, waste management has become a great public health problem for more than a decade [58]. Finding the BSF in north of Iran is promising for people involved in organic waste recycling.

BSF is known as a cosmopolitan species and has been spread across the globe for a while, however, this species was found for the first time in north of Iran. The reason why this specie

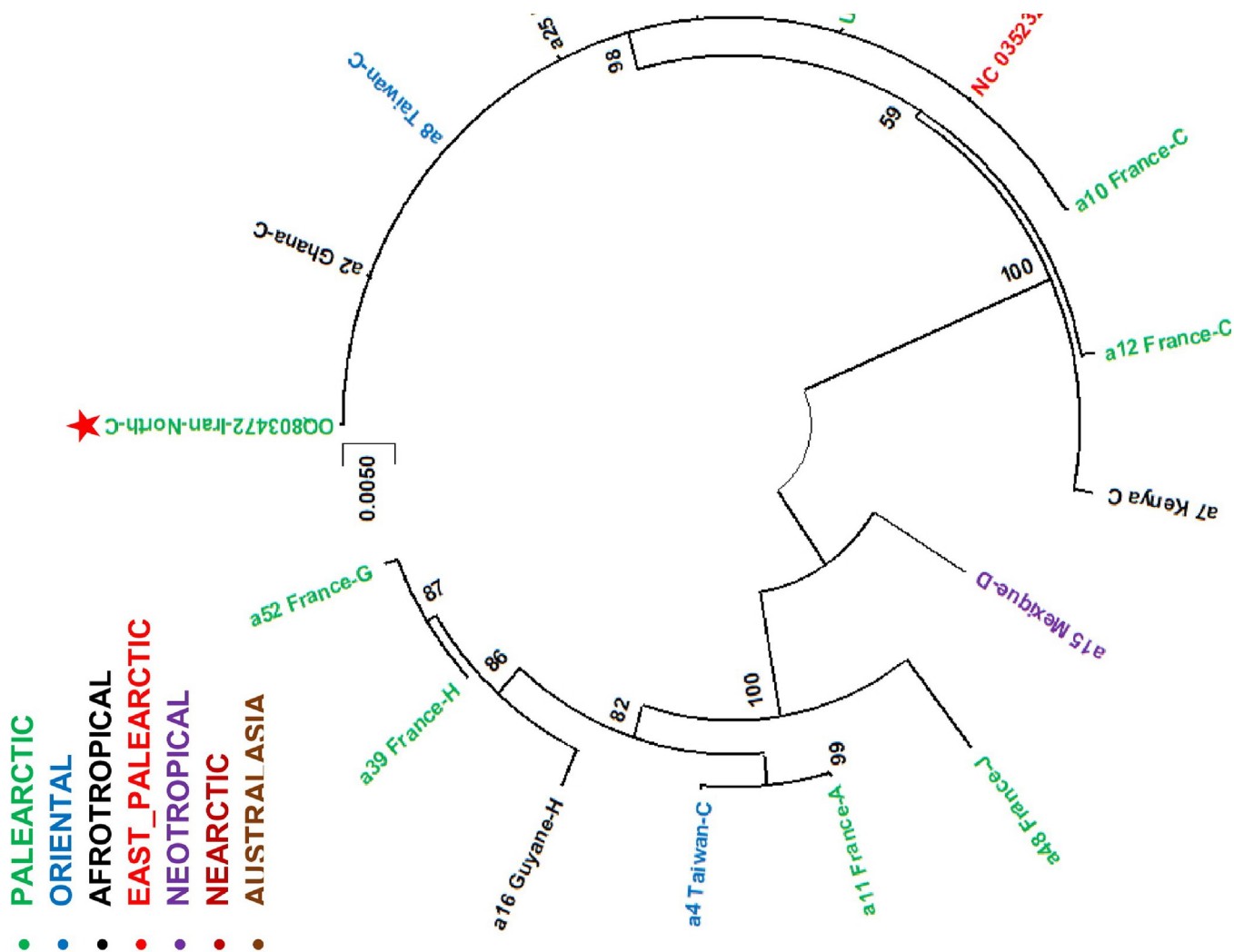

**Fig 3. The phylogenetic tree inferred by using Neighbor-Joining method based on 840 bp of the 3'-end of COI gene of *Hermetia illucens*.** The Iranian BSF is labeled with red star. The other data are representatives of available sequences in GenBank and Guilliet et al [44] and are labelled with code number, country, and haplotype (A-H) [44]. The bootstrap values are shown on the internal nodes. The colors of the branches correspond to the biogeographical regions. This tree was made by keeping only one or two sequences per haplotype except for Haplotype C.

is only now colonizing Iran, can be due to lack of comprehensive observational studies in the country. In this study we performed an initial survey for the presence of BSF on the Caspian Sea coast in northern Iran. However, a simple longitudinal observation study could determine the establishment of the species in the region.

Animal farming, particularly poultry, is one of the main activities of people in northern Iran. However, with the extension of intensive animal farming in the region, expanding amounts of animal manure demand to be processed. The long-term collection of large amounts of animal manure with lack of effective treatment can source many risks including creating more odor, producing more pathogenic microbes, making anxiety for surrounding residents, and making eutrophication of adjacent surface water bodies because of rainwater erosion, thus polluting human and farm water sources [59, 60]. The needs for treatments of livestock and poultry manure warrant investment in the rearing of BSF.

The rearing of BSF and establishing a cultured strain from this field population seems convenient in these areas, since black soldier fly larvae (BSFL) are a valuable biomass and can provide protein-rich feed for animals. The black soldier fly larvae are known to be voracious feeders that ingest and degrade from 55 up to 70% of organic materials [61, 62]. The larvae can consume most organic products and by-products, from agriculture by-products [22, 63] and manure [28, 64, 65] to animal remains [66, 67]. These wastes can then be converted into high-value insect biomass that is rich in both proteins and fats and can be used as feed for the poultry and aquaculture industries [6, 68]. Insect biomass can also reduce the requirement for certain feed nutrients (e.g., soybean) in animal farming [68]. Also, current studies have revealed the insect's ability to tolerate microplastics (MP) and the possibility is being explored that BSFL may be able to degrade polymers [69].

We found the mitochondrial haplotypes of the Iranian BSF specimens to be identical to individuals from different parts of the world including Bhutan, China, Taiwan, Russia, Netherlands, France, Spain, UK, Kenya, Ghana, and USA, of Oriental, East Palearctic, Palearctic, Afrotropical, and Nearctic biogeographic regions respectively, when either barcode or 3' end region of the COI gene (about 1500 bp totally) matched with the available data in GenBank or literature. Based on barcode region, the Iranian specimens correspond to the haplotype number one (I) as categorized by Ståhls et al. [29] and the haplotype C as categorized by Guilliet et al [52]. The latter showed that haplotype C is a commercial haplotype and includes all the sequences originated from farmed strains of the industry working on BSF. The Iranian BSF haplotype matches with what is most found in a farming context worldwide, however, there is no BSF commercial breeding site in the vicinity of the spot where the investigated individuals were captured in the wild. Finally, based on the phylogenetic analysis of this study and findings of Kaya et al. [2] we can conclude the Iranian BSF population is originated from North American strains.

Next to conservative mitochondrial genes and low/lack of intraspecific variation of ITS2 region, higher resolving molecular markers, such whole genome sequencing or microsatellites [2], could elucidate its status as being established versus occasionally present in the region, and contribute to our understanding on genotype-by-environment interactions underlying ecological adaptation and associated phenotypic variation [70]. Ecosystem sustainability is based on the balance and natural relationships of organisms and their physical environment (ecological adaptation), and the boundaries of species present are determined based on geographical and ecological constraints, the needs of each species, and the ability to spread and compete with other species. Some species enter an ecosystem other than the natural distribution range beyond their normal distribution range due to human activities (intended or unintended) and show different behaviors in the new conditions and may be able to adapt in this new habitat. Many of these species do not interfere with the new habitat and are not known as invasive species but find a niche like indigenous species. Introduced species become invasive species when they compete with indigenous species for available resources and outcompete native species from their original habitat. Humans have disrupted geographical patterns of such species by moving them around the world [71]. The probability of success of an introduced species in a new region is about one out of ten [34, 72]. Of course, one out of ten imported species can survive in nature and one out of ten established species can become a pest [34, 72]. Invasive introduced species are a major threat to biodiversity today [73]; however, since BSF is an introduced species that may lack competition with native species and can have a positive effect on cleaning the environment from toxins, organic waste, etc., it provides a wide range of usability with environmental approaches in the region while rejecting the possibility of invasion [74, 75]. However, it is not possible to determine with certainty when and where this species first established in Iran, and they have likely been released to nature due to the existence of

companies importing and breeding such flies in some provinces such as Mazandaran and Khorasan. However, not finding this species in the mentioned provinces deserves further questions and studies. One reason for not finding them in the direct vicinity of these farms may be resource availability or even adaptation of farmed BSF to specific resources not available close by. Further studies including complete genome sequencing and polymorphic nuclear genetic markers such as microsatellites could provide more information compared to the available databases [2].

Based on the present observations, this insect, originally preferring tropical climates, may exhibit cyclic occurrence in northern Iran, with a preference for cooler yet comparatively humid autumn, reiterating its immense adaptive potential to a wide range of environmental conditions. It is worth noting that the higher temperature within compost heaps may protect larvae from freezing to death and allow them to survive through a diapause-like phase [76].

Further efforts should investigate the overall presence of this fly in the country, particularly in southern Iran, close to the coasts of the Persian Gulf and Oman Sea, which have many active international trading ports, and to which the fly may already have been introduced by ships from different parts of the world.

## Conclusion

We have provided documents on the presence of BSF from North of Iran. The high genetic similarity of the Iranian BSF with commercial strains from different bio-geographical regions suggests that the extensive human-mediated translocations of the BSF have led to genetically uniform domesticated strains in various parts of the World that originated from North America. These data are important for managing and producing commercial BSF colonies to support and maintain the industries in the country that use BSF as a service provider e.g. feed, AMPs, fertilizer, and waste management processes.

## Acknowledgments

We are grateful to Dr. Martin Hauser, Senior Insect Biosystematist, from California Department of Food and Agriculture, Plant Pest Diagnostics Branch, USA for approving the species identification. We thank Mohammadreza Havasi, Fahime Rastegar, Talie Torabi Moheb and Amir Nikrooz for helping us in the field and laboratory. The manuscript was edited by the ICGEB Editing service (manuscripts@icgeb.org).

## Author Contributions

**Conceptualization:** Afrooz Boukan, Jamasb Nozari, Katayoun Pahlavan Yali, Mohammad Ali Oshaghi.

**Data curation:** Afrooz Boukan, Katayoun Pahlavan Yali, Mohammad Ali Oshaghi.

**Formal analysis:** Afrooz Boukan, Nazanin Naseri Karimi, Mohammad Ali Oshaghi.

**Funding acquisition:** Mohammad Ali Oshaghi.

**Investigation:** Afrooz Boukan, Jamasb Nozari, Nazanin Naseri Karimi, Fahimeh Talebzadeh, Katayoun Pahlavan Yali.

**Methodology:** Afrooz Boukan, Nazanin Naseri Karimi, Fahimeh Talebzadeh.

**Project administration:** Jamasb Nozari, Mohammad Ali Oshaghi.

**Resources:** Jamasb Nozari, Mohammad Ali Oshaghi.

**Software:** Afrooz Boukan, Mohammad Ali Oshaghi.

**Supervision:** Jamasb Nozari, Mohammad Ali Oshaghi.

**Validation:** Jamasb Nozari, Mohammad Ali Oshaghi.

**Visualization:** Afrooz Boukan, Nazanin Naseri Karimi, Fahimeh Talebzadeh, Katayoun Pahlavan Yali, Mohammad Ali Oshaghi.

**Writing – original draft:** Afrooz Boukan.

**Writing – review & editing:** Mohammad Ali Oshaghi.

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
