## [Decision Letter · Decision Letter 0]

5 Jul 2024

PONE-D-24-11332Mitochondrial and nuclear DNA revealed first record of Hermetia illucens (Diptera: Stratiomyidae) in Northern IranPLOS ONE

Dear Dr. Oshaghi,

Thank you for submitting your manuscript to PLOS ONE. After careful consideration, we feel that it has merit but does not fully meet PLOS ONE’s publication criteria as it currently stands. Therefore, we invite you to submit a revised version of the manuscript that addresses the points raised during the review process.

We look forward to receiving your revised manuscript.

Kind regards,

Nafiu Bala Sanda, PhD

Academic Editor

PLOS ONE

Journal Requirements:

"Tehran University of Medical Sciences"

Reviewers' comments:

Reviewer's Responses to Questions

**Comments to the Author**

1. Is the manuscript technically sound, and do the data support the conclusions?

Reviewer #1: Partly

Reviewer #2: Yes

Reviewer #3: Yes

Reviewer #4: Yes

2. Has the statistical analysis been performed appropriately and rigorously? 

Reviewer #1: N/A

Reviewer #2: N/A

Reviewer #3: Yes

Reviewer #4: N/A

3. Have the authors made all data underlying the findings in their manuscript fully available?

Reviewer #1: No

Reviewer #2: Yes

Reviewer #3: Yes

Reviewer #4: Yes

4. Is the manuscript presented in an intelligible fashion and written in standard English?

Reviewer #1: Yes

Reviewer #2: Yes

Reviewer #3: Yes

Reviewer #4: Yes

5. Review Comments to the Author

Reviewer #1: Authors said they've made all data of the study available, but only the GenBank access to the Cytochrome c Oxidase Subunit I data is given. The data for the ITS2 is missing. Results of the ITS2 are mentionned but not presented (figure).

Site of sampling the flies need to be precised (Caspian region is too large). The number of sample is never precised ("some specimens were kept alive for cultivation, and a subset of specimens was stored [...]"). Only 5 individuals were sequenced to determine the genotype of the Iranian black soldier fly, which is a very small sample size.

The abstract introduction is too general and lacks the specificity needed to effectively set the stage for the research at hand. The conclusion in the abstract should be changed since it does not relate to the subject of the research ("The results proved the presence of H. illucens in north of Iran"). Also "The introduction and

settlement of this fly in other parts of the country is highly suggested." Why ?

The introduction and discussion should be restructured ; at times the paragraph feels disjointed, with sentences that lack coherence and fail to flow seamlessly from one to the next. It reads more like a collection of facts than a cohesive narrative.

Some statements need to be revised (line 43 - BSF are not just an important source of proteins, but also of lipids as the rest of the paragraph clearly presents)(line 49 - BSF do not eat compost since the organic material as already been digested ; BSF produce compost)(line 50 - the reference propose the use of larval by-products to eradicating multidrug-resistant bacteria in aquaculture, they are not currently being used in industrial aquaculture).

In the discussion, you should add a paragraph on why this specie is only now colonizing Iran, while it has been spreed across the globe for a while. At line 229, don't you think a simple longitudinal observation study could determine the establishment of the specie in the region better than genetic markers ?

A careful re-read of the document to correct language is necessary (e.g. line 229-230 "higher resolving nuclear, such whole genome sequencing or microsatellites" or line 236-237 "Some species enter an ecosystem other than the natural distribution range beyond their normal distribution range due to [...]").

Reviewer #2: Review of PONE-D-24-11332

This is a nice article about bioinformatics study of a popular beneficial insect (Hermetia illucens). To be specific, phylogenetic investigation of Hermetia illucens in Iran. Nicely done, but there are some queries to be addressed:

1. The introduction has to be broadened in order to reach wider reader variety.

Rapid accumulation of fat in BSF must be referred in this manuscript (Biotechnology and Biofuels 12 (2019) 194 https://doi.org/10.1186/s13068-019-1531-7), because it enables BSF as a superb natural waste recycler (Cell Research 30 (2020) 50–60 https://doi.org/10.1038/s41422-019-0252-6) that opens the possibility for the transesterification of its fat as biodiesel (Sustainability 14 (2022) 13993 https://doi.org/10.3390/su142113993).

2. Moreover, the bioinformatic study of the BSF peptides should also be discussed (Scientific Reports 10 (2020) 16875 https://doi.org/10.1038/s41598-020-74017-9), as well as the gene characterization related to the energy processing in BSF, or in its mitochondria (Gene 896 (2024) 148045 https://doi.org/10.1016/j.gene.2023.148045) which are in line with this submission.

3. In addition to the phylogenetic tree in this manuscript, please also add the BLAST analysis of the gene sequences in Table 1, and hopefully a circular phylogenetic tree can be displayed in order to enrich this submission.

4. Please write scientific names in italic  Reference 6, 7, 16  Hermetia illucens

5. Reference 8: The name of authors must be revised as: van Huis A, van Itterbeeck J, Klunder H, Mertens E, Halloran A, Muir G, and Vantomme P.

6. The link for reference 8 now is https://www.fao.org/4/i3253e/i3253e.pdf

Reviewer #3: This study represents the results of genetic sequencing of mCOI1 region of black soldier flies found in the Caspian sea region of Iran. The genetic sequences were compared to other previously published sequences and available sequences on Genbank using the Neighbor-Joining method. The Iran BSF flies were found to be similar to the Haplotype C associated with reared BSF. It is unclear from the results of the timing of the introduction of the flies, or the original population due to the close relatedness present between the different sequences compared. The results represent the first reported recording of BSF in Iran and an informative paper in terms of the genetic variance and distribution of BSF around the world. The figures are clear and well presented, the materials and methods are easy to follow and support the discussion. The introduction is broad and fairly comprehensive. With some minor revisions, I would recommend this paper for publication.

Minor recommended changes and specific comments:

The short title should be amended to “Genetic Structure of Black Soldier Flies in Northern Iran”.

Abstract

Line 23 – Complete the sentence to the following “…bioconversion of organic waste into protein-rich insect larvae that can be processed into an animal feed ingredient.”

Line 28 and 29 – H. illucens and BSF are being used interchangeably here. Please amend to using one or the other for consistency and ensure that it is consistent throughout the manuscript.

Introduction

Line 43 – Replace “and” with “in”

Line 62 – Is there a museum of collected insect specimens in Iran with localities and dates that may be used to check for historical records of this species?

Materials and methods

Line 97 – Please check for consistency when referencing figures and tables throughout the manuscript (“Figure” vs “Fig.” as it is in the results section)

Line 144 – Orphaned word, is “presumably” part of the note for Table 1?

Discussion

Line 211 – remove the extra “in”

Line 212 - Larvae can be processed into a protein-rich feed ingredient, rather than a feed specifically. Please amend throughout paper where appropriate.

Line 212 – Replace “greedy” with a less emotive term such as “voracious”

Line 213 – 55 to 70% what? Please clarify here

Line 214 – - “.. consume most organic products and byproducts..” would be more appropriate than “everything”

Line 219 – “..the possibility is being explored that BSFL may be able to degrade polymers.”

Line 220 to 240 - These two sentences may have more impact at the beginning of the discussion section

Line 229 – Add “of” between lack and intraspecific

Line 236 – Add “and” after the comma

Line 244 – Unclarified statement, please provide a substantive reference for “one out of ten”

Line 247 – referring to BSF as H. illucens here is inconsistent with the rest of the manuscript

Line 248 to 250 – Please provide references for this statement, and please revise phrasing for greater clarity.

Line 251 – Replace “transferred” with “released”

Line 253 – Was there a search for the species in the named provinces for this study?

Line 250 to 253 – This sentence should be added to the abstract

Line 262 – If possible, provide a reference for this statement

Legend for Figure 2 vs Figure 3 consistency – In the figure legend the Iranian representative is referenced slightly differently, and should be changed so both figures have the same phrasing for consistency.

Reviewer #4: The present manuscript is about the Genetic structure of Hermetia illucens (Diptera: Stratiomyidae) in Northern Iran. The study is comprehensive and well structured, with some issues which need clarification.

At first, it is not very clear for me if this is the first record of the insect in Iran, or it has been recorded by other author. I found one reference as a preprint study (https://doi.org/10.21203/rs.3.rs-4293117/v1), which has to be clarified. If the present study is not about the first record, I suggest to change the title, eg. "Genetic structure of Black Soldier Fly in Northern Iran".

The abstract is well written in general and informative. Although, the sentence structure and flow could be improved for better readability.

In the introduction, consider adding more details on the ecological and economic significance of BSF, as well as some more recent literature on these. As said before, please clarify if this study is the first record of BSF in Iran or primarily focused on the use of molecular techniques for BSF taxonomy. The scope should be clearly defined.

The study area and sample collection sections are well-described. However, please include more details about the temperature and humidity, as well as coordinates. Moreover, please specify the concentration of DNA used and how it was measured. In adition the description of the phylogenetic analysis could be expanded to include more details on the software and parameters used.

In the results section, please ensure that all abbreviations are defined at first use (e.g., BSFL for black soldier fly larvae).

In the discussion I suggest to consider elaborating on the potential ecological impact of introducing BSF to new regions and discuss any limitations of the study and suggest areas for future research.

Finally, the manuscript could benefit from a proofreading for minor grammatical errors and typos.

Some specific comments/suggestions/corrections can be found in the attached pdf.

6. PLOS authors have the option to publish the peer review history of their article (what does this mean?). If published, this will include your full peer review and any attached files.

Reviewer #1: No

Reviewer #2: No

Reviewer #3: No

Reviewer #4: **Yes: **Antonios Tsagkarakis

---

## [Author Response · Author response to Decision Letter 0]

24 Jul 2024

Dear Reviewers

I should thank you for the thoughtful comments and constructive suggestions, which help to improve the quality of this manuscript. Please find in the following, point to point response to the comments.

Response to reviewer comments

Review Comments to the Author

Reviewer #1: Authors said they've made all data of the study available, but only the GenBank access to the Cytochrome c Oxidase Subunit I data is given. The data for the ITS2 is missing. Results of the ITS2 are mentionned but not presented (figure).

Response:

The respectful reviewer is quite right. We have submitted the ITS2 sequence data to GenBank and provided accession numbers which is added to the data in the manuscript. However, as explained in the manuscript, the locus was invariable across samples from diverse biogeographic regions. The locus did not resolve different populations originated from different parts of the world. So, no figure was provided.

Comment

Site of sampling the flies need to be precised (Caspian region is too large). The number of sample is never precised ("some specimens were kept alive for cultivation, and a subset of specimens was stored [...]"). 

Response:

Thanks. Sampling sites and the number of collected samples were added in the text.

Comments

Only 5 individuals were sequenced to determine the genotype of the Iranian black soldier fly, which is a very small sample size.

Response: Thanks. A valuable comment, however, unfortunately, we did not find flies in various sites to genotype different populations. We found totally 12 adult flies in a single collection site (Lahijan), and five of them were sequenced. The rest were used for colonization in the insectarium. 

Comment:

The abstract introduction is too general and lacks the specificity needed to effectively set the stage for the research at hand. The conclusion in the abstract should be changed since it does not relate to the subject of the research ("The results proved the presence of H. illucens in north of Iran"). Also "The introduction and settlement of this fly in other parts of the country is highly suggested." Why ?

Response:

The abstract introduction was modified to provide the specificity for the research. The conclusion was also modified. Due to heavy international trading in southern coast parts, the introduction and settlement of this fly in southern coast of the country is highly suggested.

Comment:

The introduction and discussion should be restructured; at times the paragraph feels disjointed, with sentences that lack coherence and fail to flow seamlessly from one to the next. It reads more like a collection of facts than a cohesive narrative.

Response

The sections were modified to provide coherence and to flow seamlessly.

Comment:

Some statements need to be revised (line 43 - BSF are not just an important source of proteins, but also of lipids as the rest of the paragraph clearly presents)(line 49 - BSF do not eat compost since the organic material as already been digested ; BSF produce compost)(line 50 - the reference propose the use of larval by-products to eradicating multidrug-resistant bacteria in aquaculture, they are not currently being used in industrial aquaculture).

Response

Thank you very much for valuable comments. The statements were revised accordingly.

Comment

In the discussion, you should add a paragraph on why this specie is only now colonizing Iran, while it has been spreed across the globe for a while. At line 229, don't you think a simple longitudinal observation study could determine the establishment of the specie in the region better than genetic markers?

Response

Thank you very much for valuable comments. A few sentences were added in the discussion section to explain the issue. 

Comment

A careful re-read of the document to correct language is necessary (e.g. line 229-230 "higher resolving nuclear, such whole genome sequencing or microsatellites" or line 236-237 "Some species enter an ecosystem other than the natural distribution range beyond their normal distribution range due to [...]").

Response

We tried to check language thoroughly. 

Reviewer #2: Review of PONE-D-24-11332

This is a nice article about bioinformatics study of a popular beneficial insect (Hermetia illucens). To be specific, phylogenetic investigation of Hermetia illucens in Iran. Nicely done, but there are some queries to be addressed:

1. The introduction has to be broadened in order to reach wider reader variety.

Rapid accumulation of fat in BSF must be referred in this manuscript (Biotechnology and Biofuels 12 (2019) 194 https://doi.org/10.1186/s13068-019-1531-7), because it enables BSF as a superb natural waste recycler (Cell Research 30 (2020) 50–60 https://doi.org/10.1038/s41422-019-0252-6) that opens the possibility for the transesterification of its fat as biodiesel (Sustainability 14 (2022) 13993 https://doi.org/10.3390/su142113993).

Response

Thanks for valuable comment. The introduction has been broadened and the suggested references have been added accordingly.

2. Moreover, the bioinformatic study of the BSF peptides should also be discussed (Scientific Reports 10 (2020) 16875 https://doi.org/10.1038/s41598-020-74017-9), as well as the gene characterization related to the energy processing in BSF, or in its mitochondria (Gene 896 (2024) 148045 https://doi.org/10.1016/j.gene.2023.148045) which are in line with this submission.

Response

Thanks for valuable comment. The introduction has been modified and the suggested references have been added accordingly.

3. In addition to the phylogenetic tree in this manuscript, please also add the BLAST analysis of the gene sequences in Table 1, and hopefully a circular phylogenetic tree can be displayed in order to enrich this submission.

Response

Thanks for valuable comment. The result of Blast search was added in the legend of the table. Also a circular phylogenetic tree for both loci were provided.

4. Please write scientific names in italic  Reference 6, 7, 16  Hermetia illucens

Response

Thanks. The scientific names were written italic.

5. Reference 8: The name of authors must be revised as: van Huis A, van Itterbeeck J, Klunder H, Mertens E, Halloran A, Muir G, and Vantomme P.

6. The link for reference 8 now is https://www.fao.org/4/i3253e/i3253e.pdf

Response

Thanks. The reference was corrected accordingly.

Reviewer #3: This study represents the results of genetic sequencing of mCOI1 region of black soldier flies found in the Caspian sea region of Iran. The genetic sequences were compared to other previously published sequences and available sequences on Genbank using the Neighbor-Joining method. The Iran BSF flies were found to be similar to the Haplotype C associated with reared BSF. It is unclear from the results of the timing of the introduction of the flies, or the original population due to the close relatedness present between the different sequences compared. The results represent the first reported recording of BSF in Iran and an informative paper in terms of the genetic variance and distribution of BSF around the world. The figures are clear and well presented, the materials and methods are easy to follow and support the discussion. The introduction is broad and fairly comprehensive. With some minor revisions, I would recommend this paper for publication.

Response: Thanks for your positive assessment.

Minor recommended changes and specific comments:

The short title should be amended to “Genetic Structure of Black Soldier Flies in Northern Iran

Response:

The title has been modified accordingly.

Abstract

Line 23 – Complete the sentence to the following “…bioconversion of organic waste into protein-rich insect larvae that can be processed into an animal feed ingredient.”

Response.

The sentence has been completed accordingly.

Line 28 and 29 – H. illucens and BSF are being used interchangeably here. Please amend to using one or the other for consistency and ensure that it is consistent throughout the manuscript.

Response: The manuscript has been modified accordingly.

Introduction

Line 43 – Replace “and” with “in”

Response: The manuscript has been modified accordingly.

Line 62 – Is there a museum of collected insect specimens in Iran with localities and dates that may be used to check for historical records of this species?

Response: Yes, there are several insect museum in Iran, including two important ones, one in the University of Tehran and one in the University of Medical Sciences. Some specimens have been deposited in the Tehran University of Medical Sciences. We have added the following sentence in the 1st paragraph of result section. Also a few voucher specimens were deposited in the Insect Museum of School of Public Health, Tehran University of Medical Sciences.

Materials and methods

Line 97 – Please check for consistency when referencing figures and tables throughout the manuscript (“Figure” vs “Fig.” as it is in the results section)

Response: The manuscript has been checked and modified accordingly.

Line 144 – Orphaned word is “presumably” part of the note for Table 1?

Response: The word was deleted.

Discussion

Line 211 – remove the extra “in”

Response: The extra in was deleted.

Line 212 - Larvae can be processed into a protein-rich feed ingredient, rather than a feed specifically. Please amend throughout paper where appropriate.

Response: The text was amended throughout where appropriate.

Line 212 – Replace “greedy” with a less emotive term such as “voracious”

Response: The manuscript has been changed accordingly.

Line 213 – 55 to 70% what? Please clarify here

Response: The manuscript has been changed accordingly.

Line 214 – - “.. consume most organic products and byproducts..” would be more appropriate than “everything”

Response: The manuscript has been changed accordingly.

Line 219 – “..the possibility is being explored that BSFL may be able to degrade polymers.”

Response: The manuscript has been changed accordingly.

Line 220 to 240 - These two sentences may have more impact at the beginning of the discussion section

Response: The two sentences have been moved to the beginning of discussion section.

Line 229 – Add “of” between lack and intraspecific

Response: “of” was added accordingly.

Line 236 – Add “and” after the comma

Response: “and” was added accordingly.

Line 244 – Unclarified statement, please provide a substantive reference for “one out of ten”

Response: Two references were added accordingly.

Line 247 – referring to BSF as H. illucens here is inconsistent with the rest of the manuscript

Response: H. illucens was deleted and replaced with BSF.

Line 248 to 250 – Please provide references for this statement, and please revise phrasing for greater clarity.

Response: Two references were added accordingly.

Line 251 – Replace “transferred” with “released”

Response: “transferred” was replaced with “released”

Line 253 – Was there a search for the species in the named provinces for this study?

Response: Not yet.

Line 250 to 253 – This sentence should be added to the abstract

Response: The sentence was added to the abstract.

Line 262 – If possible, provide a reference for this statement

Response: A reference was added accordingly.

Legend for Figure 2 vs Figure 3 consistency – In the figure legend the Iranian representative is referenced slightly differently, and should be changed so both figures have the same phrasing for consistency.

Response: The figure legends were homogenized accordingly.

Reviewer #4: The present manuscript is about the Genetic structure of Hermetia illucens (Diptera: Stratiomyidae) in Northern Iran. The study is comprehensive and well structured, with some issues which need clarification.

At first, it is not very clear for me if this is the first record of the insect in Iran, or it has been recorded by other author. I found one reference as a preprint study (https://doi.org/10.21203/rs.3.rs-4293117/v1), which has to be clarified. If the present study is not about the first record, I suggest to change the title, eg. "Genetic structure of Black Soldier Fly in Northern Iran".

Response:

Thanks for your valuable comment. At the time of submission of this manuscript, there was no record on the presence of this species in Iran. However, we have changed the manuscript and deleted “the first report”. The tile also was changed to the suggestion of respectful reviewer.

The abstract is well written in general and informative. Although, the sentence structure and flow could be improved for better readability.

Response:

We have tried to modify the structure.

In the introduction, consider adding more details on the ecological and economic significance of BSF, as well as some more recent literature on these. As said before, please clarify if this study is the first record of BSF in Iran or primarily focused on the use of molecular techniques for BSF taxonomy. The scope should be clearly defined.

Response:

We have modified the text and explained that our report is the second report for the country in the discussion section. 

The study area and sample collection sections are well-described. However, please include more details about the temperature and humidity, as well as coordinates. Moreover, please specify the concentration of DNA used and how it was measured. In adition the description of the phylogenetic analysis could be expanded to include more details on the software and parameters used.

Response:

We have shown the temperature and humidity of the collection site as well as coordinates in the text. The quality and quantity of the extracted DNA was estimated using the spectrophotometer and gel electrophoresis. The amount of DNA for PCR was added. And the details of phylogenetic analysis was added in the text.

In the results section, please ensure that all abbreviations are defined at first use (e.g., BSFL for black soldier fly larvae).

Response:

All the abbreviations were checked.

In the discussion I suggest to consider elaborating on the potential ecological impact of introducing BSF to new regions and discuss any limitations of the study and suggest areas for future research.

Response:

Thanks. All the issues were considered and the text modified accordingly.

Finally, the manuscript could benefit from a proofreading for minor grammatical errors and typos.

Response:

Thanks. We have tried our best to reduce the grammatical errors and typos. 

Some specific comments/suggestions/corrections can be found in the attached pdf.

Response:

Thanks. Thanks for your kind help and offering valuable comments. All the comments were considered positively, and correction were made accordingly. Unfortunately, I did not get your comment on the 1st and 2nd lines of pdf file. However, based on your comments here, we did corrections.

Best regards

---

## [Editor Report · Decision Letter 1]

29 Jul 2024

Genetic Structure of Black Soldier Flies in Northern Iran

PONE-D-24-11332R1

Dear Dr. Mohammad Ali Oshaghi,

We’re pleased to inform you that your manuscript has been judged scientifically suitable for publication and will be formally accepted for publication once it meets all outstanding technical requirements.

Kind regards,

Nafiu Bala Sanda, PhD

Academic Editor

PLOS ONE

Additional Editor Comments (optional):

The revised manuscript can be accepted for publication in PLOS ONE journal, Congratulations!
---

## [Editor Report · Acceptance letter]

5 Aug 2024

PONE-D-24-11332R1 

PLOS ONE

Dear Dr. Oshaghi, 

I'm pleased to inform you that your manuscript has been deemed suitable for publication in PLOS ONE. Congratulations! Your manuscript is now being handed over to our production team.

Kind regards, 

on behalf of

Dr. Nafiu Bala Sanda 

Academic Editor

PLOS ONE